# Identification and Quantification of Bioactive Compounds in Organic and Conventional Edible Pansy Flowers (*Viola × wittrockiana*) and Their Antioxidant Activity

**DOI:** 10.3390/plants12061264

**Published:** 2023-03-10

**Authors:** Michalina Kozicka, Ewelina Hallmann

**Affiliations:** 1Institute of Human Nutrition Sciences, Department of Functional and Organic Food, Warsaw University of Life Sciences, Nowoursynowska 159c, 02-776 Warsaw, Poland; 2Bioeconomy Research Institute, Agriculture Academy, Vytautas Magnus University, K. Donelaičio Str. 58, 44248 Kanuas, Lithuania

**Keywords:** anthocyanins, carotenoids, conventional, edible flowers, organic, polyphenols

## Abstract

The use of edible flowers has become increasingly popular as a good source of bioactive compounds. Many flowers can be consumed, but there is a lack of information about the chemical composition of organic and conventional flowers. Organic crops represent a higher level of food safety because pesticides and artificial fertilizers are prohibited. The present experiment was carried out with organic and conventional edible pansy flowers of different colors: double-pigmented violet/yellow and single-pigmented yellow flowers. In fresh flowers, the contents of dry matter and polyphenols (including phenolic acids, flavonoids, anthocyanins, carotenoids, and chlorophylls) and the antioxidant activity were determined by the HPLC-DAD method. The results showed that organic edible pansy flowers contained significantly more bioactive compounds, especially polyphenols (333.8 mg/100 g F.W.), phenolic acids (40.1 mg/100 g F.W.), and anthocyanins (293.7 mg/100 g F.W.) compared to conventional methods. Double-pigmented (violet/yellow) pansy flowers are more recommended for the daily diet than single-pigmented yellow flowers. The results are unique and open the first chapter in a book on the nutritional value of organic and conventional edible flowers.

## 1. Introduction

In human nutrition, fruits and vegetables are commonly used as food resources. Typical plant parts with consumable properties include stems (cauliflowers, kohlrabi), leaves (lettuce, cabbage), fruits (tomato, cucumber), seeds (beans, peas), and roots (carrot, red beetroot) [1,2,3,4,5]. For plants, flowers are mostly used as organs for reproduction and fruit and seed production [6]. Only a few species have edible flowers: broccoli, baby broccoli, flowering cabbage, and artichokes [7,8,9]. Fruits and vegetables are perfect sources of vitamins and bioactive compounds such as polyphenols (flavonoids, anthocyanins) and carotenoids [10,11,12]. Since they are strong antioxidants, these compounds have beneficial effects on human health [13,14]. As reported in the latest literature, many noninfective chronic diseases start from excessive concentrations of free radicals and oxidative stress. The use of fruits and vegetables rich in antioxidants can diminish oxidative stress by neutralizing free radicals, which can lead to decreases in chronic diseases such as many types of cancer, obesity, diabetes, and neurodegenerative problems such as Parkinson’s and Alzheimer’s diseases [15,16,17,18,19]. Flowers are produced by humans mainly as ornamental plants [20]. The garden pansy (*Viola × wittrockiana*) is a type of large-flowered hybrid plant cultivated as a garden flower [21]. The genus *Viola* divides all cultivated varieties (cultivars) into four subgroups: B1—pansies, B2—violas, B3—violettas, and B4—cornuta hybrids. On this classification, modern “pansies” differ from the other three subgroups by possessing a well-defined “blotch” or “eye” in the middle of the flower. Modern horticulturalists tend to use the term “pansy” for those multi-colored, large-flowered hybrids that are grown for bedding purposes every year, while “viola” is usually reserved for smaller, more delicate annuals and perennials [22]. In the past, viola flowers were cultivated in private gardens in the 18th century. Modern horticulturists have developed a wide range of pansy flower colors and double colors, including yellow, gold, orange, purple, violet, red, white, and even near-black (very dark purple). Pansies typically display large, showy face markings. Plants grow well in sunny or partially sunny positions in well-draining soils. Pansies are perennials but are normally grown as biennials or annuals because of their leggy growth [23]. For better growth, they are watered thoroughly about once a week, depending on climate and rainfall. The plant should never be over-watered. To maximize blooming, plant fertilization is used about every other week [24].

Many experiments have shown that many flowers can be treated as edible plant parts. It is well known that more than 80 flower species have edible flowers [25,26,27,28]. Edible flowers can combine a few scopes: nutritional, decorative, and pro-healthy properties such as antioxidants in chronic disease prevention and oxidative stress diminishing [29,30]. Flowers contain numerous bioactive compounds, such as carotenoids (orange and red colorants), xanthophylls and flavonoids (yellow colorants), anthocyanins (purple, pink, and red colorants), and chlorophylls (green colorants), therefore they have such beautiful colors [31,32,33].

Due to environmental biotic and abiotic stresses, organic plant management involves higher bioactive compound synthesis in plants [34,35,36]. This leads to higher concentrations of antioxidants in fruits and vegetables. Many experiments have shown that organic fruits, vegetables, and medicinal herbs are characterized by a higher antioxidant status [37,38,39]. Of course, there are contrary results as well [40,41]. In the present literature, there are no data about the comparison of bioactive value between organic and conventional edible flowers. Many experiments show only the chemical composition of conventional edible flowers [42,43,44]. The aim of this work is to show the differences in chemical composition and bioactive compound concentrations between organic and conventional edible flowers of the pansy (*Viola × wittrociana*) featuring different colors.

## 2. Results and Discussion

The obtained results showed that organic pansy flowers were characterized by a significantly higher (*p* < 0.0001) concentration of dry matter than conventional flowers (Table 1a).

A higher concentration of dry matter is characteristic of organic crops and practices. This is connected to the “water swelling” phenomenon. The tissues of conventional plants collect more water than those of organic plants because plants absorb a large amount of water together with mineral fertilizers [45]. Many scientific studies have shown similar results among organic and conventional fruits and vegetables [46,47,48,49,50,51], but it is possible to find opposite results [52,53,54]. The authors did not observe statistical differences in dry matter content between different pigmented pansy flowers. Pansy flowers contained 14.5 g/100 g F.W. and 14.2 g/100 g F.W. of dry matter in the violet/yellow and yellow flowers, respectively. Flowers with deep colors have the potential to produce higher dry matter concentrations. The black locus (with flowers) was characterized by lower dry matter content than the bristly locus (pink flowers) [55]. A similar relationship was observed with daisy flowers. Pink daisy flower species contained 18.63 g/100 g F.W. and white daisy flowers contained 15.97 g/100 g F.W. of dry matter [56]. The interaction between production systems and flower pigmentations indicated that flowers from organic production with both colors (yellow and violet/yellows) were characterized by a higher and more significant (*p* < 0.0001) dry matter content in comparison to conventional flowers (Table 1b).

According to the data presented in Table 1a, organic pansy flowers contained significantly (*p* < 0.0001) more polyphenol compounds than conventional flowers, at 333.8 mg/100 g F.W. and 245.1 mg/100 g F.W., respectively. Since the use of pesticides is prohibited in organic agriculture, plants start to produce bioactive compounds from polyphenolic groups to protect themselves from pests and diseases [57]. Plants that are not chemically protected react with an increased synthesis of polyphenolic compounds compared to conventional plants that are cultivated with pesticides. This is one of the plant stress reactions [58]. Only a small portion of the polyphenols were phenolic acids. Organic pansy flowers contained 2.8-fold more phenolic acids than conventional flowers (Table 1a). Double-colored pansy flowers produced significantly more polyphenol (*p* < 0.0001) compounds than single-colored yellow pansy flowers, at 519.9 mg/100 g F.W. and 58.9 mg/100 g F.W. Similar findings were noted in experiments with 22 different pigmented pansy flowers. Violet/yellow flowers contained 661.0 mg/100 g F.W. polyphenols, while light yellow flowers contained only 73.0 mg/100 g F.W. polyphenols [59]. The authors observed a strong interaction between the production system and flower pigmentation. As presented in Table 1b, organic violet/yellow and yellow pansy flowers contained significantly more polyphenols. It is a confirmed theory that in organic systems, plants react similarly to environmental conditions and agricultural practices, increasing phenolic synthesis in both violet/yellow and yellow pansy flowers. No experiments with phenolic acids and organically edible pansy flowers were found, but similar results were obtained for pomegranate fruits. The values for polyphenols were 2270 mg/100 g F.W. and 1651 mg/100 g F.W. for organic and conventional pomegranate juices, respectively [60]. Pansy flowers with a yellow color were characterized by a significantly higher concentration of phenolic acids compared to conventional flowers, at 35.5 mg/100 g F.W. and 20.6 mg/100 g F.W. Phenolic acids are colorless or light-colored chemical compounds. Similar findings were presented with edible flowers with light and dark tones. White daisy flowers, as well as yellow cosmos flowers, contain more phenolic acids than flowers with much deeper pigmentation, such as dark red rose flowers and blue clitoria flowers [61]. In the conventional system, violet/yellow pansy flowers contained significantly more (*p* < 0.0001) phenolic acids compared to yellow ones. On the other hand, in an organic system, we observed a contrary situation because yellow pansy flowers were characterized by a higher concentration of phenolic acids (Table 1b).

In the presented experiment, four individual phenolic acids (gallic, chlorogenic, caffeic, and *p*-coumaric) were identified in pansy flowers. These phenolic acids are characteristic of *Viola* species [47,48]. The concentrations of all phenolic acids were significantly higher in organic flowers (Table 2a). Similar results were reported for organic cabbage and carrots but not for red beetroots [49,62,63]. Double-pigmented pansy flowers contained significantly more caffeic and *p*-coumaric acids than single-pigmented yellow pansy flowers (Table 2a). Bristly locust and red horse-chestnut flowers with dark (pink) petal colors were characterized by higher contents of phenolic acids such as caffeic acid, gallic acid, or chlorogenic acid compared to those with white flowers (black locust and horse chestnut) [55,64]. In the case of violet/yellow pansy flowers, we observed a significant interaction of experimental factors only in *p*-coumaric acid concertation. In both systems (organic and conventional), double-pigmented pansy flowers contained significantly (*p* < 0.0001) more *p*-coumaric acid. Organic cultivation systems is much better for the yellow pansy flowers. However, conventional cultivation systems are better for violet/yellow pansy flowers, since it increases the concentrations of gallic and chlorogenic acids in the flowers (Table 2b).

Flavonoids, in addition to their functions to help plants adapt to their surrounding environment, have also been correlated with numerous health benefits in humans, including antioxidant, antimicrobial, and anticancer activities [29,31]. Organic pansy flowers contained significantly more flavonoids than conventional flowers, with flavonoid contents of 293.7 mg/100 g F.W. and 229.1 mg/100 g F.W. (Table 1a). There were no similar results, so the concentration of flavonoids in pansy flowers was compared with that of strawberry and sour cherry fruits. As reported in the literature, organic fruits are characterized by a higher concentration of flavonoids than conventional fruits [65,66]. The authors have much more information on the differences in the contents of flavonoid compounds in pansy flowers with different petal pigmentations. Violet/yellow pansy flowers contained 577.4 mg/100 g F.W. up to 670.4 mg/100 g F.W. [59]. In contrast, bright yellow pansy flowers had only 8.62 mg/100 g F.W. flavonoids [67]. Another experiment showed that *Viola × wittrockiana* flowers (light yellow flowers) contained 21.5 mg/100 g F.W. of total flavonoids [68]. The obtained results confirmed this information in the literature. In terms of total flavonoids, violet/yellow pansy flowers contained 499.3 mg/100 g F.W., and yellow flowers contained 23.5 mg/100 g F.W. (Table 1a). The violet color of pansy flowers is given by anthocyanins, which belong to flavonoids. In the presented experiment, the authors observed a strong interaction between production systems and flower pigmentation. Both double-pigmented and yellow pansy flowers contained significantly more flavonoids (*p* < 0.0001) in an organic system compared to a conventional one (Table 1b). In the case of flowers with violet pigmentation, more flavonoids were obtained compared to one-pigmented flowers. In organic flowers, there was a higher concentration of bioactive compounds (Table 1b). In the present experiment, five individual flavonoids (quercetin-3-*O*-rutinoside, myricetin, quercetin, quercetin-3-*O*-glucoside, and kaempferol) were identified and quantified in pansy flowers. These flavonoid compounds are characteristic of pansy flowers. This was confirmed in other studies where quercetin and its derivatives, as well as kaempferol, were identified [69]. In the group of individual flavonoids, only kaempferol occurred in higher concentrations in conventional pansy flowers. The remaining flavonoids (quercetin and its derivatives, and myricetin) were abundant at higher concentrations in organic pansy flowers (Table 2a). Similar relationships were presented for organic and conventional sour cherries and raspberries [39,66]. Pansy flowers with violet/yellow pigmentation were characterized by higher concentrations of quercetin-3-*O*-rutinoside, quercetin-3-*O*-glucoside, and pure quercetin than yellow pansy flowers. A similar relationship was discussed in another experiment with wild pansies (*Viola cornuta*) with light-pigmented flowers and cultivated *Viola × wittrockiana* with violet/yellow flowers. Pansies with darker flowers contained 10.1 mg/100 g F.W. of quercetin-3-*O*-rutinoside, while those with lighter pigmentation contained only 7.5 mg/100 g F.W. In the case of kaempferol, *Viola × wittrockiana* contained 0.75 mg/100 g F.W., and *Viola cornuta* contained 0.71 mg/100 g F.W. [70]. The obtained results confirm these findings. Pansy with violet/yellow flowers contained 0.40 mg/100 g F.W. of quercetin-3-*O*-rutinoside, and yellow flowers contained only 0.29 mg/100 g F.W. (Table 2). A strong interaction between production systems and flower colors was found. Double-pigmented pansy flowers in both systems (organic and conventional) were characterized by a higher and more significant (*p* < 0.0001) concentration of quercetin derivates (Table 2b). The concentrations of mirycetin and quercetin were higher in conventional violet/yellow pansies, as well as in organic yellow pansies (Table 2b).

Anthocyanins and their derivatives are water-soluble flavonoids and natural pigments that are responsible for the color of flowers. Their color depends mainly on pH, but metal ions and co-pigments may also affect pH. These flavonoids are responsible mainly for the red, blue, and purple colors of flowers. Anthocyanins and their color in flowers play a significant role in plants since they are responsible for correct pollination, and the color of flowers is necessary to attract pollinators (birds and insects). In addition, for humans, anthocyanins have been correlated with plants with increased antioxidant activity and therefore with high nutritional value [33]. A high concentration of anthocyanins in plants is an effect of their ability to absorb UV radiation in the environment [71]. Pansy flowers with deep purple and violet colors are perfect sources of different anthocyanins. According to the data presented in Table 1a, organic pansy flowers contained significantly more total anthocyanins (241.9 mg/100 g F.W.) compared to conventional pansy flowers (209.9 mg/100 g F.W.). There were no similar results in the literature. Authors can compare these results to organic dark-pigmented fruits such as blackcurrant or blue honeysuckle berries from organic production [72,73]. Organic fruits, similar to organic flowers, contain significantly more total anthocyanins than conventional fruits. Pansy flowers with deep violet pigmentation were characterized by the highest concentration of anthocyanins (451.9 mg/100 g F.W.). However, anthocyanins were not detected in yellow pansy flowers. Similar findings were presented by others: violet/yellow pansy flowers contained 242.5 mg/100 g F.W. and yellow only traces of total anthocyanins [59]. Violet pansy flowers contained less than 555.7 mg/100 g F.W. of total anthocyanins, while yellow anthocyanins were much less abundant than 19.9 mg/100 g F.W. [74]. Only two individual anthocyanins were identified and qualified in pansy flowers: cyanidin-3-*O*-rutinoside and cyanidin-3-*O*-glucoside (Table 2a). Similar results were obtained by others [75,76]. In the petals of pansy flowers, the dominant anthocyanin compound is cyanidin-3-*O*-rutinoside. Organic foods with violet/yellow flowers contained significantly more (*p* < 0.0001) of this anthocyanin compound compared to conventional ones. In the case of the second purple pigment, cyanidin-3-*O*-glucoside—conventional violet/yellow pansy flowers contained significantly more of that colorant (Table 2b).

Carotenoids, including xanthophylles and carotenes, are lipophilic pigments that accumulate in flowers and are responsible for color that attracts pollinators. Flower petals have a very wide range of carotenoid levels, depending on the plant species or cultivar [77,78]. The total carotenoid content is presented in Table 1a. Organic pansy flowers were characterized by a significantly higher concentration of total carotenoids than conventional flowers (8.5 mg/100 g F.W. and 5.3 mg/100 g F.W., respectively). Since there is a lack of information about the carotenoid contents of organic and conventional flowers, it was possible to compare the results obtained only for orange and yellow-pigmented fruits. According to data published by others, organic orange and persimmon fruits from organic production contained more total carotenoids than conventional fruits [79,80]. Pansy flowers with one-colored petals also contained more carotenoids than those with violet stripes. Similar results were presented in other experiments. White, yellow, and red pansy flowers contained 3.1 mg/100 g F.W., 8.3 mg/100 g F.W., and 15.6 mg/100 g F.W. of total carotenoids, respectively [81]. In the presented experiment, violet/yellow pansy flowers contained 9.6 mg/100 g F.W., and yellow flowers contained 4.2 mg/100 g F.W. (Table 1a). As we demonstrated, pansy flowers reacted positively to production in experimental production systems. On the basis of the obtained results, we conclude that for yellow-pigmented pansy flowers, cultivation in organic systems is much better, because organic one-pigmented pansy flowers were characterized by a significant (*p* = 0.0001) and higher (11.28 mg/100 g F.W. and 7.88 mg/100 g F.W., respectively) concentration of total carotenoids compared to conventional flowers (Table 1b).

In the experiment conducted, organic pansy flowers were characterized by significantly higher concentrations of all identified carotenoids (Table 2a). Organic farm management stimulates plants to produce fruits with a higher β-carotene content. Organic apricots contained 1.4–2.4 mg/100 g F.W. of β-carotene, while conventional apricots contained 1.5–1.7 mg/100 g F.W. [82,83]. The yellow color in pansy flowers is created by xanthophylls and carotenes. In two xanthophylls (lutein and zeaxanthin), one carotene compound has been identified and quantified. Carotenoids in flower petals make flowers more attractive for pollination. Some insects react only to two types of light wavelengths because they have two types of eye receptors. One absorbs green and yellow light, and the other absorbs blue and ultraviolet light. However, a large proportion of insects have trichromatic vision, e.g., honeybees, bumblebees, and many types of butterflies, which means that they have three receptors that allow them to perceive almost full colors, with absorption maxima of ultraviolet, blue-violet, and yellow pigmentation. Xanthophylls and carotenes absorb visible light in the range of 445–470 nm, which is why the examined pansy flowers with a higher proportion of yellow color in the petals contained more carotenoids [81]. Yellow pansy flowers were characterized by a significantly higher β-carotene concentration than conventional flowers (1.7 mg/100 g F.W. and 1.4 mg/100 g F.W.) (Table 2b) [81]. The all-yellow pansy flowers contained 6.8 mg/100 g F.W. of lutein, while double-pigmented (violet/yellow) flowers contained only 1.7 mg/100 g F.W. Similar findings were presented in another experiment: yellow pansy flowers contained 1.3 mg/100 g F.W. lutein and red ones only 0.1 mg/100 g F.W. [82,83,84]. A strong and significant interaction between cultivation and flower pigmentation was found. Organic yellow pansy flowers were characterized by a higher concentration of lutein (*p* = 0.0043) and β-carotene (*p* = 0.004) compared to conventional flowers. Only in the case of zeaxanthin did we find contrary results, because conventional violet/yellow pansy flowers and organic yellow pansy flowers were characterized by a higher concentration of that yellow pigment (0.98 mg/100 g F.W. and 1.16 mg/100 g F.W., respectively) (Table 2b).

Chlorophyll in pansy flowers is a green pigment occurring in the stems and base of the flower (perianth leaves). The authors noted a significant difference between organic and conventional pansy flowers in total chlorophyll content (Table 1a), but not in the case of differentiation according to flower pigmentation. Yellow and double-colored pansy flowers contained similar concentrations of total chlorophyll. The obtained results have been confirmed by other experiments [85]. A strong and significant interaction (*p* = 0.003) was found between production systems (organic and conventional) as well as flower color (Table 1b). The authors observed that yellow pansy flowers, both organic and conventional, were characterized by a higher concentration of total chlorophyll (9.37 mg/100 g F.W. and 14.57 mg/100 g F.W., respectively), compared to conventional ones (Table 1b).

The chlorophyll a and b contents in pansy flowers were 5.3 mg/100 g F.W. for yellow pansy flowers, 8.3 mg/100 g F.W. for violet/yellow flowers for chlorophyll b, and 6.7 mg/100 g F.W. and 2.8 mg/100 g F.W. for chlorophyll a. Similar results were presented in another experiment with different colors of pansy flowers (3.0–4.7 mg/100 g F.W. for chlorophyll b and 4.8–8.0 mg/100 g F.W. for chlorophyll a) [85]. In the case of chlorophyll b, both organic and conventional pansy flowers were characterized by a significant (*p* = 0.0001) and higher concentration of that pigment (6.09 mg/100 g F.W. and 10.52 mg/100 g F.W., respectively). In the case of chlorophyll a, a contrary situation was observed. In both systems, more chlorophyll a was contained in yellow pansy flowers (5.86 mg/100 g F.W. and 7.54 mg/100 g F.W., respectively) (Table 2b).

According to the data presented in Figure 1, double-pigmented pansy flowers (from both production systems) were characterized by a higher level of antioxidant activity than yellow pansy flowers, which were mostly associated with anthocyanin occurrence. These compounds are characterized by a high level of antioxidant activity compared to carotenoids. It is worth noting that organic pansy flowers were characterized by significantly higher antioxidant activity (190.7 µmol TEAC/100 g F.W. and 99.4 µMol TEAC/100 g F.W.) compared to conventional TEAC (182.0 µMol TEAC/100 g F.W. and 67.9 µmol TEAC/100 g F.W.). Similar results, not with pansy flowers, but with different pigmented corn cultivars, showed that corn with violet kernels was characterized by higher antioxidant activity (3.93 µMol TEAC/g D.W.) compared to yellow ones (1.39 µMol TEAC/g D.W.) [86]. In another experiment, potato with yellow flesh was characterized by a lower antioxidant activity (71 µMol TEAC/100 g D.W.), due to only carotenoids occurring, compared to purple/yellow potato flesh with mostly anthocyanins (299 µMol TEAC/100 g D.W.) [87]. Bioactive compounds identified and quantified in organic and conventional pansy flower petals play very important roles as antioxidant agents that can protect plants from environmental stresses. In the case of humans, edible flowers can be treated similarly to fruits and vegetables as important elements of antioxidant protection in the body as well as dietary elements. In the experiment, strong correlations between polyphenols and antioxidant activity were obtained in both organic and conventional pansy flowers (Table 3). In the violet/yellow pansy flower petals, antioxidant activity depended mostly on anthocyanin content (R^2^ = 0.872). In the yellow fruit compounds responsible for antioxidant activity, carotenoids (R^2^ = 0.955 and R^2^ = 0.909) were present in organic and conventional pansy flowers, respectively (Table 3). Results obtained by others have been confirmed by the presented experiment, where double-pigmented pansy flower antioxidant activity was higher than that of single-pigmented orange snapdragon flowers [69]. Yellow single-pigmented pumpkin flowers were characterized by a lower antioxidant power compared to double-pigmented violet/yellow pansy flowers [47].

## 3. Materials and Methods

### 3.1. Chemical Reagents

The following chemical reagents were used in this study: 2′2-azinebis-3-ethylbenzothiazolin-6-sulfonic acid (Merck, Warsaw, Poland), acetic acid (Chempur, Warsaw, Poland), acetonitrile (Sigma-Aldrich, Warsaw, Poland), deionized water (Sigma-Aldrich, Warsaw, Poland), ethyl acetate (Sigma-Aldrich, Warsaw, Poland), hydrochloric acid (Chempur, Warsaw, Poland), magnesium carbonate (Sigma-Aldrich, Warsaw, Poland), methanol (Merck, Warsaw, Poland), ortho-phosphoric acid (Chempur, Warsaw, Poland), and phenolic compound standards, including gallic acid (CAS 149-91-7), chlorogenic acid (CAS 327-97-9), *p*-coumaric (CAS 501-98-4), caffeic acid (CAS 331-39-5), quercetin-3-*O*-glucoside (CAS 482-35-9), quercetin-3-*O*-rutinoside (CAS 207671-50-9), myrycetin (CAS 529-44-2), quercetin (CAS 849061-97-8), cyanidin-3-*O*-glucoside (CAS 7084-24-4), and cyjanidin-3-*O*-rutinoside (CAS 18719-76-1), lutein (CAS 127-40-2), zeaxanthin (CAS 144-68-3), β-carotene (CAS 7235-40-7), chlorophyll a (CAS 479-61-8), chlorophyll b (CAS 519-62-0) (Sigma-Aldrich, Supelco, Poland), phosphate buffered saline, and potassium persulfate (Sigma-Aldrich, Warsaw, Poland).

### 3.2. Flower Origins

Experiments were carried out in May 2022. Pansy flowers were collected separately on an organic farm located in Rozalin (52°27′79″ N; 17°95′19″ E) and a conventional farm located in Podkowa Leśna (52°07′18″ N; 20°43′35″ E). Flowers were collected in the morning and quickly transported to a chemical laboratory. The transport was extremely delicate in order to avoid the destruction of the flowers. Single flowers were put into paper (made from soft cellulose) envelopes and put into a thermal box with dry ice inside. To avoid the damaging low temperature action on flower petals, soft cellulose and a layer of polystyrene were used as a border between dry ice. Every sample was represented by fifty flowers. For analysis purposes, two different varieties were collected: violet/yellow pansy flowers and full yellow pansy flowers (Figure 2A,D). Fresh flowers were divided into flower petals and stems. For experimental purposes, only petals with perianth leaves were used. Each sample (according to flower color) was divided into two parts. One part was used only for dry matter evaluation. The second part was freeze-dried using a Labconco (2.5) freeze-dryer (Warsaw, Poland, −40 °C, pressure 0.100 mbar). After the freeze-drying process, the experimental material was ground in a laboratory mill (A11). Then, the ground samples were stored at –80°C in small scyntylic tubes in an ultra-low freezer (U501 Labit) to avoid the loss of bioactive compounds.

### 3.3. Analysis of Dry Matter

The dry matter content of pansy flowers was measured before the freeze-drying process. The dry matter content was determined using a scale as described in Polish Norm PN-R-04013:1988 [88]. Flower samples were dried at 105 °C for 48 h using an FP-25 W Farma Play dryer (Bytom, Poland). The dry matter content was calculated for the pansy flower samples based on their mass differences and is given in units of g/100 g F.W. (fresh weight).

### 3.4. Analysis of Polyphenols (Flavonoids and Phenolic Acids)

Polyphenols (flavonols and phenolic acids) were measured using the HPLC-DAD [89]. The first step of extraction was weighing 100 mg of freeze-dried powdered flower tissue, mixed with 5 mL of 80% methanol (HPLC grade), on a Vortex 326 M (Marki, Poland). Furthermore, all samples were extracted in an ultrasonic bath (10 min, 30 °C, 5.5 kHz). After 10 min of extraction, the flower samples were moved to a centrifuge (10 min, 6000 rpm, 5 °C). After centrifugation, each supernatant was collected into clean Eppendorf tubes and centrifuged again (5 min, 12,000 rpm, 0 °C). A total of 500 μL of supernatant was transferred to HPLC vials and analyzed. For polyphenol separation and identification, a Synergi Fusion-RP 80i Column 250 × 4.60 mm (Phenomenex, Warsaw, Poland) was used. Analysis was conducted with the use of Shimadzu equipment (Ilinois, USA), including two LC-20AD pumps, a CBM-20A controller, an SIL-20AC column oven, and UV/Vis SPD-20 AV and SPD-M20A spectrometers. For the separation of phenolic compounds (flavonols and phenolic acids), gradient conditions with a flow rate of 1 mL/min were used. Two gradient phases were used: 10% (v:v) acetonitrile and ultrapure water (phase A), and 55% (v:v) acetonitrile and ultrapure water (phase B). The phases were acidified with orthophosphoric acid (pH 3.0). The total time of the analysis was 38 min. The phase-time program was as follows: 1.00–22.99 min, 95% phase A and 5% phase B; 23.00–27.99 min, 50% phase A and 50% phase B; 28.00–28.99 min, 80% phase A and 20% phase B; and 29.00–38.00 min, 95% phase A and 5% phase B. The wavelengths of detection were 250 nm for phenolic acids and 370 nm for flavonols. Identification and quantification of flavonoids and phenolic acids were performed according to pure chemical standards and spectra (Appendix A). The content of individual flavonoids and phenolic acids was calculated on the basis of standard curve equations (detailed standard equations and graphs are given in Appendix A).

### 3.5. Analysis of Polyphenols (Anthocyanins)

The first step of anthocyanin extraction was flavonoid and phenolic acid extraction. The prepared supernatant, in a volume of 2.5 mL, was mixed with 2.5 mL of 10 M hydrochloric acid (HCl) and 100% methanol (HPLC grade) in 5 mL. Samples were collected in a refrigerator (10 min, 5 °C). Next, 1 mL of the extract was transferred to HPLC vials and used for analysis by the HPLC-DAD method according to a previously described method [89]. Anthocyanins were separated under isocratic conditions with a flow rate of 1.5 mL/min. The single mobile phase contained acetic acid (5%), methanol (HPLC-pure), and acetonitrile (HPLC-pure) (70:10:20). The analysis time was 10 min, with detection at 570 nm. The identification and quantification of anthocyanins were performed according to standards and spectra (Appendix A). The content of individual anthocyanins was calculated on the basis of standard curve equations (detailed standard equations and graphs are given in Appendix A).

### 3.6. Analysis of Carotenoids

Carotenoids and chlorophylls were measured by HPLC [90]. Fifty milligrams of freeze-dried flower powdered tissue was mixed with 5 mL of 100% acetone (HPLC grade) and magnesium carbonate (1 mg) and mixed on a Vortex 326 M (Marki, Poland). Then, all samples were extracted in a cold ultrasonic bath (15 min, 0 °C, 5.5 kHz). After 15 min of extraction, the samples were transferred to a centrifuge (15 min, 5000 rpm, 0 °C). After centrifugation, each supernatant was collected in a clean Eppendorf tube and centrifuged again (5 min, 12,000 rpm, 0 °C). A total of 900 μL of supernatant was transferred to HPLC vials and analyzed using a 50-µL column injection. For carotenoid and chlorophyll separation and identification, an Ma-RP 80i Column 250 × 4.60 mm (Phenomenex, Warsaw, Poland) was used. The analysis was conducted with Shimadzu equipment, as described in previous subsections. For the separation of carotenoids and chlorophyll compounds, gradient conditions of two phases with a flow rate of 1 mL/min were used: acetonitrile with 90:10 methanol (phase A) and methanol with 68:32 ethyl acetate (phase B). The total time of the analysis was 25 min. The phase-time program was as follows: 1.00–14.99 min, 100% phase A; 15.00–22.99 min, 40% phase A and 60% phase B; 23.00–27.99 min, 100% phase B. The wavelengths for detection were 445 nm for xanthophylls and 450 nm for carotenes and chlorophylls. Identification and quantification of carotenoids and chlorophylls were performed according to standards and spectra (Appendix A). The contents of individual carotenoids and chlorophylls were calculated on the basis of standard curve equations (detailed standard equations and graphs are given in Appendix A).

### 3.7. Analysis of Antioxidant Activity

A total of 265 mg of potassium persulfate (K_2_S_2_O_8_) was dissolved in 20 mL of distilled water. Five milliliters of distilled water followed by 5 mL of a previously prepared aqueous solution of potassium persulfate were added to 384 mg of the ABTS^•+^ (2′2-azinebis-3-ethylbenzothiazolin-6-sulfonic acid) reagent. The solution was prepared a minimum of 12 h before the planned assay and stored in a dark, cool place. A total of 250 mg of the freeze-dried plant material was weighed into a plastic tube with a cap (50 mL), and 25 mL of distilled water was added. The tube was then placed on a vortex shaker (LP shaker Vortex, Labo Plus, Warsaw, Poland) for 60 s at 2000 rpm for complete mixing. Subsequently, the sample was incubated in a shaker incubator (IKA KS 4000 Control, IKA, Staufen im Breisgau, Germany) for 60 min (30 °C, 6000 rpm). After incubation, the sample was again shaken on a vortex shaker for 60 s to ensure complete mixing, and then centrifuged (MPW-380 R, Warsaw, Poland) at 5 °C and 8000 for 20 min. After centrifugation, the supernatant was used for determinations. In 10-mL glass tubes, test extract solution, measured with a predetermined dilution scheme (0.5–1.5 mL), was then added to 3.0 mL of ABTS^•+^ cationic solution in PBS (phosphate-buffered saline). Absorbance measurements were taken exactly 6 min after incubation at room temperature. Absorbance was measured at a wavelength λ = 734 nm using a spectrophotometer (Helios, Thermo Scientific, Warsaw, Poland). The obtained measurements were calculated using a special formula (y = −5.6017x + 0.7134) including the dilution factor. The results were expressed as mmol of TE (Trolox equivalents per 100 g F.W. (fresh weight of flowers)) [91].

### 3.8. Statistical Analysis

The obtained results were statistically elaborated. For experimental purposes, Statgraphics Centurion 15.2.11.0 software (StatPoint Technologies, Inc., Warranton, VA, USA) was used. The statistical calculations were based on a one-way analysis of variance with the use of Tukey’s test (*p* = 0.05). The experimental factors were the origin and production of pansy flowers (organic and conventional) and the color of the flowers (bicolored violet/yellow flowers) and one-color flowers (yellow). A lack of statistically significant differences between the examined groups is indicated by similar letters. The standard error (SE) is provided with each mean value reported in the tables. The sum of polyphenols was significantly higher in organic pansy flowers.

## 4. Conclusions

The presented research shows that organic pansy flowers were characterized by high contents of biologically active compounds compared to conventional flowers. Two-colored violet/yellow flowers were much richer in bioactive compounds than single-colored flowers. It is due to the high presence of anthocyanins with strong antioxidative properties. It seems that modification of cultivation systems can influence the composition of bioactive compounds in edible pansy flowers. Conventional agriculture is preferable for double-pigmented pansy flowers, while organic systems are much better for one-colored (yellow) pansy flowers. Based on the presented experiment, organic edible pansy flowers could be recommended for consumption as a great source of antioxidant compounds.

## Figures and Tables

**Figure 1 plants-12-01264-f001:**
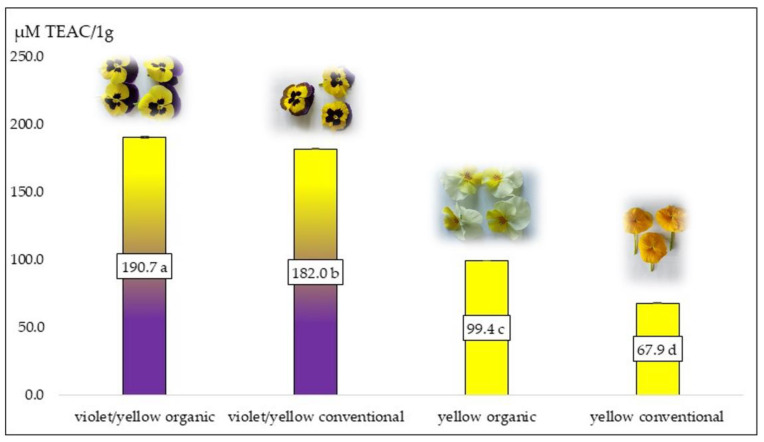
Antioxidant activity of different pansy flowers. Means on bars followed by the same letter are not significantly different at the 5% level of probability (*p* < 0.05).

**Figure 2 plants-12-01264-f002:**
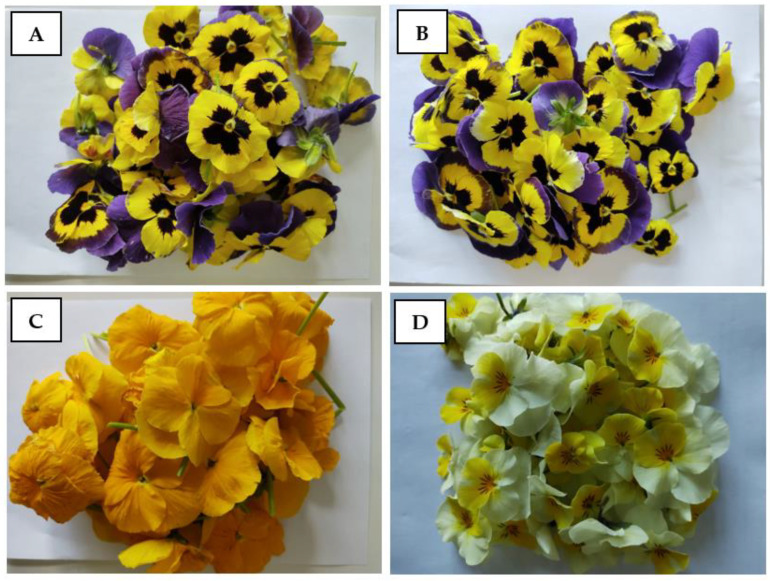
Conventional double-pigmented pansy flowers: yellow/violet (**A**); organic double-pigmented, yellow/violet pansy flowers (**B**); conventional, one-pigmented, yellow pansy flowers (**C**); and organic, one-pigmented, yellow pansy flowers (**D**).

**Table 1 plants-12-01264-t001:** (**a**) The mean value for groups of different bioactive compounds (counted sum) and the content of dry matter (in g/100 g F.W.) in pansy flowers with different flower colors and production systems (in mg/100 g F.W.). (**b**) The interaction between experimental factors for groups of different bioactive compounds (counted sum) and the content of dry matter (in g/100 g F.W.) in pansy flowers with different flower colors and production systems (in mg/100 g F.W.).

(**a**)						
**Compound Groups/Experimental Combinations**	**Production System**		**Flower Color**		***p*-Value**	
	**Conventional**	**Organic**	**(Violet/Yellow)**	**(Yellow)**	**Production System**	**Flower Color**
dry matter	13.40 ± 0.4 ^1^ B	15.32 ± 0.4 A	14.49 ± 0.1 a	14.24 ± 0.8 a	<0.0001	N.S. ^3^
polyphenols	245.06 ± 87.4 ^1^ B ^2^	333.76 ± 100.9 A	519.87 ± 25.1 a	58.95 ± 11.4 b	<0.0001	<0.0001
phenolic acids	16.00 ± 0.9 B	40.07 ± 6.9 A	20.58 ± 1.1 b	35.50 ± 8.8 a	<0.0001	<0.0001
flavonoids	229.06 ± 86.5 B	293.69 ± 107.8 A	499.29 ± 24.1 a	23.46 ± 2.6 b	<0.0001	<0.0001
anthocyanins	209.95 ± 85.7 B	241.95 ± 98.8 A	451.90 ± 13.4	not detected	<0.0001	
carotenoids	5.29 ± 1.1 B	8.46 ± 1.2 A	4.17 ± 0.6 b	9.58 ± 0.7 a	<0.0001	<0.0001
chlorophylls	9.16 ± 0.2 B	13.89 ± 0.4 A	11.08 ± 0.9 a	11.97 ± 1.1 a	<0.0001	N.S.
(**b**)						
**Compound Groups/Experimental Combinations**	**Conventional**		**Organic**		***p*-Value**	
	**Violet/Yellow Pansy**	**Yellow Pansy**	**Violet/Yellow Pansy**	**Yellow Pansy**		
dry matter	14.48 ± 0.13 ^1^ b ^2^	12.33 ± 0.13 c	14.50 ± 0.10 b	16.15 ± 0.28 a	<0.0001	
polyphenols	458.96 ± 3.97 a	31.16 ± 0.61 c	580.78 ± 6.22 a	86.74 ± 1.60 b	<0.0001	
phenolic acids	18.03 ± 0.21 c	13.98 ± 0.42 d	23.12 ± 0.35 b	57.02 ± 1.09 a	<0.0001	
flavonoids	440.93 ± 3.53 b	17.19 ± 0.28 c	557.65 ± 5.93 a	29.73 ± 0.54 c	<0.0001	
anthocyanins	419.91 ± 3.48 b	not detected	483.90 ± 4.27 a	not detected	<0.0001	
carotenoids	2.69 ± 0.02 c	7.88 ± 0.20 b	5.65 ± 0.06 b	11.28 ± 0.29 a	0.0001	
chlorophylls	8.95 ± 0.31 b	9.37 ± 0.26 b	13.21 ± 0.32 a	14.57 ± 0.40 a	0.003	

^1^ Data are presented as the mean ± SE (standard error) with an ANOVA *p*-value. ^2^ Means in a row followed by the same letter are not significantly different at the 5% level of probability (*p* < 0.05); ^3^ Not significant statistically.

**Table 2 plants-12-01264-t002:** (**a**) The mean values for the content of dry matter (in g/100 g F.W.) and other bioactive compounds (in mg/100 g F.W.) in pansy flowers with different flower colors and flowers from different production systems. (**b**) The interaction between experimental factors and the content of individual and identified bioactive compounds (in mg/100 g F.W.) in pansy flowers with different flower colors and from different production systems.

**(a)**						
**Compounds/Experimental Combinations**	**Production System**		**Flowers’ Color**		***p*-Value**	
	**Conventional Pansy**	**Organic Pansy**	**(Violet/Yellow)**	**(Yellow)**	**Production System**	**Flower Color**
gallic	8.79 ± 0.6 B	23.29 ± 5.3 A	10.27 ± 0.2 b	21.82 ± 5.9 a	<0.0001	<0.0001
chlorogenic	4.03 ± 0.1 B	10.57 ± 1.8 A	5.15 ± 0.4 b	9.45 ± 2.3 a	<0.0001	<0.0001
caffeic	0.86 ± 0.1 B	1.05 ± 0.1 A	1.03 ± 0.1 a	0.88 ± 0.1 b	0.0091	0.0003
*p*-coumaric	2.32 ± 0.2 B	5.16 ± 0.2 A	4.13 ± 0.6 a	3.35 ± 0.6 b	<0.0001	<0.0001
quercetin-3-*O*-rutinoside	0.24 ± 0.1 B	0.45 ± 0.1 A	0.40 ± 0.1 a	0.29 ± 0.1 b	<0.0001	<0.0001
myticetin	3.92 ± 0.2 B	6.07 ± 0.1 A	5.21 ± 0.3 a	4.78 ± 0.5 b	<0.0001	0.0124
quercetin	1.83 ± 0.1 B	2.32 ± 0.1 A	2.18 ± 0.1 a	1.98 ± 0.1 b	<0.0001	0.0005
quercetin-3-*O*-glucoside	10.93 ± 0.5 B	40.80 ± 9.0 A	37.41 ± 10.4 a	14.32 ± 1.8 b	<0.0001	<0.0001
kaempferol	2.19 ± 0.1 A	2.10 ± 0.1 B	2.20 ± 0.1 a	2.09 ± 0.1 b	0.0155	0.004
cyanidin-3-*O*-rutinoside	184.33 ± 75.3 B	217.02 ± 88.6 A	401.35 ± 13.6	not detected	<0.0001	<0.0001
cyanidin-3-*O*-glucoside	25.62 ± 10.5 A	24.93 ± 10.2 B	50.56 ± 0.5	not detected	<0.0001	<0.0001
lutein	3.01 ± 1.0 B	5.61 ± 1.1 A	1.76 ± 0.5 b	6.87 ± 0.6 a	<0.0001	<0.0001
zeaxanthin	0.94 ± 0.1 B	1.08 ± 0.1 A	0.99 ± 0.1 b	1.02 ± 0.1 a	<0.0001	0.048
β-carotene	1.34 ± 0.1 B	1.77 ± 0.1 A	1.43 ± 0.1 b	1.69 ± 0.1 a	<0.0001	0.0001
chlorophylls b	4.80 ± 0.6 B	8.78 ± 0.7 A	8.30 ± 0.9 a	5.27 ± 0.7 b	<0.0001	<0.0001
chlorophylls a	4.36 ± 0.6 B	5.12 ± 1.0 A	2.78 ± 0.1 b	6.70 ± 0.4 a	0.0106	<0.0001
**(b)**						
**Compounds Groups/Experimental Combinations**	**Conventional**		**Organic**		***p*-Value**	
	**Violet/Yellow Pansy**	**Yellow Pansy**	**Violet/Yellow Pansy**	**Yellow Pansy**		
gallic	10.17 ± 0.16 ^1^ b ^2^	7.42 ± 0.30 c	10.36 ± 0.30 b	36.22 ± 0.63 a	<0.0001	
chlorogenic	4.20 ± 0.06 c	3.85 ± 0.14 c	6.10 ± 0.03 b	15.05 ± 0.34 a	<0.0001	
caffeic	0.96 ± 0.01	0.76 ± 0.01	1.10 ± 0.02	1.00 ± 0.03	N.S. ^3^	
*p*-coumaric	2.70 ± 0.01 b	1.94 ± 0.03 b	5.57 ± 0.07 a	4.75 ± 0.15 a	0.0005	
quercetin-3-*O*-rutinoside	0.27 ± 0.01 b	0.20 ± 0.01 b	0.52 ± 0.01 a	0.37 ± 0.01a	0.0003	
myrycetin	4.38 ± 0.06 b	3.47 ± 0.03 b	6.05 ± 0.10 a	6.09 ± 0.18 a	0.0077	
quercetin	2.05 ± 0.03 a	1.62 ± 0.01 b	2.30 ± 0.02 a	2.34 ± 0.05 a	0.0002	
quercetin-3-glucoside	11.99 ± 0.10 b	9.86 ± 0.26 b	62.82 ± 1.62 a	18.79 ± 0.31 b	<0.0001	
kaempferol	2.34 ± 0.02	2.03 ± 0.02	2.06 ± 0.01	2.14 ± 0.04	N.S.	
cyanidin-3-*O*-rutinodise	368.66 ± 2.92 b	not detected	434.04 ± 3.86 a	not detected	<0.0001	
cyanidin-3-*O*-glucoside	51.25 ± 0.61 a	not detected	49.86 ± 0.41 b	not detected	<0.0001	
lutein	0.57 ± 0.02 c	5.45 ± 0.17a	2.94 ± 0.05 b	8.29 ± 0.25 a	0.0043	
zeaxanthin	0.98 ± 0.01 a	0.89 ± 0.01 b	0.99 ± 0.01 a	1.16 ± 0.02 a	<0.0001	
β-carotene	1.14 ± 0.03 b	1.54 ± 0.03 a	1.71 ± 0.03 a	1.83 ± 0.03 a	0.004	
chlorophyll b	6.09 ± 0.35 b	3.51 ± 0.15 c	10.52 ± 0.36 a	7.04 ± 0.16 a	0.0001	
chlorophyll a	2.86 ± 0.05 b	5.86 ± 0.21 a	2.70 ± 0.15 b	7.54 ± 0.27 a	0.0037	

^1^ Data are presented as the mean ± SE (standard error) with an ANOVA *p*-value. ^2^ Means in a row followed by the same letter are not significantly different at the 5% level of probability (*p* < 0.05); ^3^ N.S., statistically not significant.

**Table 3 plants-12-01264-t003:** Pearson’s coefficient (R^2^) between antioxidant activity and total polyphenols, anthocyanins, and carotenoids in organic and conventional pansy flowers with different colors.

Groups of Compounds	Antioxidant Activity			
	Organic Pansy Flowers	*p*-Value	Conventional Pansy Flowers	*p*-Value
total polyphenols	0.9997	<0.0001	0.9998	<0.0001
	violet/yellow		yellow	
total anthocyanins	0.8726	<0.0001	-	
total carotenoids	0.9553	<0.0001	0.9095	<0.0001

## Data Availability

Not applicable.

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
