# Peer review of "Identification and Quantification of Bioactive Compounds in Organic and Conventional Edible Pansy Flowers (Viola × wittrockiana) and Their Antioxidant Activity"

_plants, 2023, doi:10.3390/plants12061264_

Round 1
Reviewer 1 Report (Previous Reviewer 2)
Thank you for allowing me to review this article of great scientific interest again. The authors have made changes in the text that have contributed to improving the quality of the publication. I want to thank them for their effort in considering all the proposed comments. I congratulate them for the great work done. Therefore, the article is now suitable to be published in Plants.
Author Response
Reply to Reviewer no. 1
Comment 1: “Thank you for allowing me to review this article of great scientific interest again. The authors have made changes in the text that have contributed to improving the quality of the publication. I want to thank them for their effort in considering all the proposed comments. I congratulate them for the great work done. Therefore, the article is now suitable to be published in Plants.”
Authors’ response: We are grateful to Reviewer no. 1 for they hard work. Thank you very much for the review and for your positive recommendation to publish our manuscript in the ‘Plants’ journal.
Reviewer 2 Report (Previous Reviewer 1)
Dear Author,
After peer review of the manuscript “Plants-2268925” entitled “Identification and quantification of bioactive compounds in or- 2 ganic and conventional edible pansy flowers (Viola × wittrock- 3 iana) and their antioxidant activity” it was found to be a well-designed study of edible pansy flowers towards identification/quantification of elite chemical components as well as their antioxidant property evaluation. However, the manuscript is not acceptable in its present format. The decision over the manuscript is “Major Revision”. All the required corrections are highlighted inside the manuscript in yellow color with attached comment boxes. Authors are asked to go through all of them and correct them.
Comments:
1. Abstract: Line number 24: Avoid use of the words such as- I, We, Our, your etc.
2. Keywords:Remove the word "pansy" from keywords.
3. Introduction:Page 2, Line number 45-46: The statement is grammatically wrong. Re-write the whole statement.
4. Page 2, Line number 62-64: According to reference number [29], it has three different prospectives, right?? But I think it has more than only three prospectives... Author need to find out some more and should add in the introduction section.
5. Page 2, Line number 64-65: Its not a perfect statement. Just because it has color, it constitutes numerous bioactive compounds. Its not the case always.
6. Page 2, Line number 68-69: What is the difference between organic cultivation and the plants or crops that grows by itself in wild or in its natural habitat??
Because, I think in both the cases environmental biotic and abiotic stresses will be similar. Then how can author say that in "organic plant management bioactive compounds synthesis will be high??"
7. Page 2, Line number 72-77: This is okay. But the research gaps and novelty of the study is not properly established in the Introduction section. Author need to add one para reviewing the need of the study, and depicting the novelty of the study in the present day scenario as well as with its future approaches.
8. Results and Discussion: Page 4, Line number 115-116: This statement needs a perfect updated reference.
9. Page 4, Line number 117-118: There is no link between these two statements. Why?? Line number.
10. Page 4, Line number 128: Grammatical mistake. "It is confirm theory..."
11. Page 4, Line number 131: I want to see the results of reference number [63]. Spo that it will be easy for the readers to compare.
12. Page 4, Line number 135: What are "Withe daisy flowers???" Is it white daisy flowers or what?? Need clarifications in this aspect.
13. Page 5, Line number 176-178: Where is the full stop "." for this statement??
14. Page 5, Line number 178: "In our experiment".... You should write "In the present experiment".
15. Page 6, Line number 213-214: What is conventional treatment??
16. Table: Table 2a: Are these words complete??? Something is missing, I think.
17. Table 2a: Instead of writing "beta" author should use symbols. Follow the pattern throughout the whole manuscript.
18. Page 10, Line number 353-355: It should not be like "Our results have been confirmed by others"... It should be your approach that you have confirmed and validated previously published results from this present investigation. Author needs to rewrite the statement with scientifically sound manner.
19. Figure 1: Author should compare this antioxidant activity with a standard antioxidative agent as well.
20. Materials and Methods: Page 12, Line number 409-410: Grammatical mistake.
21. Page 12, Line number 411-412: By whom exactly the identification was carried out?? Need to mention the name with designation of the identifier.
22. Page 12, line number 412: What did the author meant by "extragently"??
23. Page 12, Line number 423: What was the instrument used to store the samples at -80 Degree C??
24. Page 14, Line number 530: Will it really be "resolved"??
25. Page 14, Line number 547-548: Provide the formula in equation form.
26. Conclusions: Page 15, Line number 563: Avoid the use of words such as our, we, I, you etc., throughout the manuscript.
27. Page 15, Line number 566-567: Statement is grammatically wrong.
28. Page 15, Line number 570-571: This cannot be recommended for every one right. Because toxicological site of consumption is still need to be evaluated.
29. References: All the references must be in accordance to the Journal format.
30. Additionally, the language of the manuscript must be revised with the help of a language expert. The revised version of the manuscript should must be devoid of any grammatical or typical errors.

Author Response
Reviewer no. 2
Thank you very much for the review and for your positive recommendation to publish our manuscript in the ‘Plants’ journal. Below you can find answers for your comments and suggestions:
Comment 1: “Abstract: Line number 24: Avoid use of the words such as- I, We, Our, your etc.”
Authors’ response: Authors agree with Reviewer suggestion. Author's personification is unnecessary in scientific articles. All such errors have been corrected in the text of the whole manuscript.”
Comment 2: “ Keywords:Remove the word "pansy" from keywords”
Authors’ response: Authors remove word pansy from keywords
Comment 3: “Introduction:Page 2, Line number 45-46: The statement is grammatically wrong. Re-write the whole statement”
Authors’ response: Authors want to apologise. Whole statement was corrected in manuscript text.
Comment 4: “Page 2, Line number 62-64: According to reference number [29], it has three different prospectives, right?? But I think it has more than only three prospectives... Author need to find out some more and should add in the introduction section. ”
Authors’ response: In those lines Authors give only few (three) selected examples of pansy properties given: nutritional, decorative and antioxidant properties. Of course, there are much more of them. Others properties were added to manuscript text with relevant references.
Comment 5: “ Page 2, Line number 64-65: Its not a perfect statement. Just because it has color, it constitutes numerous bioactive compounds. Its not the case always.”
Authors’ response: The Authors agree with the Reviewer suggestion. The sentence has been revised to make it more relevant and understandable.
Comment 6: “Page 2, Line number 68-69: What is the difference between organic cultivation and the plants or crops that grows by itself in wild or in its natural habitat?? ” Because, I think in both the cases environmental biotic and abiotic stresses will be similar. Then how can author say that in "organic plant management bioactive compounds synthesis will be high??"
Authors’ response: Authors want to answer for this question, but there is a long and the complicated explanation. First, the principles of organic farming consist in administering appropriate organic fertilizers (plants’ and animals’). Wild growing plants cannot use fertilizers. They only have organic matter available in the soil, but it can be uptake completely unbalanced. In organic farming, the amount of introduced macronutrients and microelements is controlled. Plants growing wild in the environment do not have such opportunities. Next, cultivated plants in organic system they have natural pest management with extracts and plant macerates. In natural conditions, wild plants are left without chemical protection and plant prevention. In organic plant cultivation, system plants are regularly watered. Wild plants obtain water only during normal rainfall. In organic farming competitive plants, known as weeds, are eliminated by mechanical removal. Nobody does that in nature. Plants compete with weeds for nutrients and water. Here you see only examples of differences between organic plant management and wild plants growth conditions. In organic plant cultivation, conditions similar to natural ones are created by applying the previously mentioned activities. Organic farming systems and wild growing conditions are therefore very similar, especially in terms of natural (biotic and abiotic stresses). There are many studies showing that plants in the organic system produce more bioactive compounds. Numerous earlier publications presented by the Authors as well show these differences. Many of the Authors' earlier articles cannot be cited in this work, although many of them are shown here. In the past Authors conducted a many experiments in type "form farms to fork", showing the differences between organic and conventional crops management as well they effect on human health. The results of previous experiments and articles in many scientific journals give to Authors such possibilities to claim, that: “organic plant management influence on the bioactive compounds synthesis in plants.”
Refenences:
- Średnicka-Tober D., Barański M., Gromadzka-Ostrowska J., Skwarło-Sońta K., Rembiałkowska E., Hajšlová J., Schulzová V., Cakmak I., Królikowski T., Hallmann E., Baca E., Eyre M., Steinshamn H., Jordon T., Leifert C. Effect of crop protection and fertilization regimes used in organic and conventional production systems on feed composition and physiological parameters in rats, Journal of Agricultural and Food Chemistry, 2013, 61, 5, p. 1017-1029;
- Kazimierczak R., Hallmann E., Lipowski J., Drela N., Kowalik A., Püssa T., Matt D., Luik A., Gozdowski D., Rembiałkowska E. Beetroot (Beta vulgaris L.) and naturally fermented beetroot juices from organic and conventional production: metabolomics, antioxidant levels and anticancer activity, Journal of the Science of Food and Agriculture, 2014, 94, 3, p. 2618-2629
- Kazimierczak R., Hallmann E., Rembiałkowska E. Effects of organic and conventional production systems on the content of bioactive substances in four species of medicinal plants, Biological Agriculture & Horticulture, 2015, 31, p. 118-127
- Hallmann E., Kazimierczak R., Marszałek K., Drela N., Kiernozek E., Toomik P., Matt D., Luik A., Rembiałkowska E. The nutritive value of organic and conventional white cabbage (Brassica oleracea var. capitata) and anti-apoptotic activity in gastric adenocarcinoma cells of sauerkraut juice produced therof, Journal of Agricultural and Food Chemistry, 2017, 65, p. 8171-8183
- Hallmann E., Marszałek K., Lipowski J., Jasińska U., Kazimierczak R., Średnicka - Tober D., Rembiałkowska E. Polyphenols and carotenoids in pickled bell pepper from organic and conventional production, Food Chemistry, 2018, 278, p. 254-260
- Carrillo C., Wilches-Pérez D., Hallmann E., Kazimierczak R., Rembiałkowska E. Organic versus conventional beetroot. Bioactive compounds and antioxidant properties, LWT-Food Science and Technology, 2019, 116, p. 1-7
- Hallmann E., Rozpara E., Słowianek M., Leszczyńska J. The effect of organic and conventional farm management on the allergenic potency and bioactive compounds status of apricots (Prunus armeniaca L.), Food Chemistry, 2019, 279, p. 171-178.
- Ponder A., Hallmann E. The effects of organic and conventional farm management and harvest time on the polyphenol content in different raspberry cultivars, Food Chemistry, 2019, 301, p. 171-178.
- Ponder A., Hallmann E. Phenolics and carotenoid contents in the leaves of different organic and conventional raspberry (Rubus idaeus L.) cultivars and their in vitro activity, Antioxidants, 2019, 8, p. 1-13
- Kazimierczak R., Średnicka- Tober D., Hallmann E., Kopczyńska K., Zarzynska K. The impact of organic vs. conventional agricultural practices on selected quality features of eight potato cultivars, Agronomy, 2019, 9, p. 1-15
- Kopczyńska K., Kazimierczak R., Średnicka-Tober D., Szafirowska, A., Barański, M., Rembiałkowska E., Hallmann E. The effect of organic vs. conventional cropping system on the yield and chemical composition of three courgette cultivars, Agronomy-Basel, 2020, 10, 9, 1-19.
- Lasinskas M., Jariene E., Vaitkeviciene N., Kulaitiene J., Najman K., Hallmann E. Studies of the variability of polyphenols and carotenoids in different methods fermented organic leaves of willowherb (Chamerion angustifolium (L.) Holub), Applied Sciences, 2020, 10, 15, p. 1-1
- Najman K., Sadowska A., Hallmann E. Influence of thermal processing on the bioactive, antioxidant, and physicochemical properties of conventional and organic agriculture black garlic (Allium sativum L.), Applied Sciences, 2020, 10, 23, p. 1-17.
- Aninowski M., Kazimierczak R., Hallmann E., Rachtan-Janicka J., Fijoł-Adach E., Feledyn-Szewczyk B., Majak I., Leszczyńska J. Evaluation of the potential allergenicity of strawberries in response to different farming practices, Metabolites, 2020, 10, 3, p. 1-16.
- Vaitkeviciene N., Kulaitiené J., Jariene E., Levickiene D., Danillčenko H., Średnicka - Tober D., Rembiałkowska E., Hallmann E. Characterization of bioactive compounds in colored potato (Solanum tuberosum) cultivars grown with conventional, organic, and biodynamic methods, Sustainability, 2020, 12, 7, p. 1-13
- Głowacka A., Rozpara E., Hallmann E. The dynamic of polyphenols concentrations in organic and conventional sour cherry fruits: Results of a 4-year field study, Molecules, 2020, 25, 16, p. 1-12.
- Hallmann E., Ponder A., Aninowski M., Naragerel T., Leszczyńska J. The interaction between antioxidants content and allergenic potency of different raspberry cultivars, Antioxidants, 2020, 9, 256, 1-15
- Hallmann E., Sabała P. Organic and conventional herbs quality reflected by their antioxidant compounds concentration, Applied Sciences-Basel, 2020,10, 10, 1-11.
- Lasinskas M., Jariene E., Vaitkeviciene N., Blinstrubiene A., Sawicka B., Sadowska A., Hallmann E.: Studies of the variability of sugars, vitamin c, and chlorophylls in differently fermented organic leaves of Willowherb (Chamerion angustifolium (L.) Holub), Applied Sciences-Basel, 2021, 11, 21, 1-8.
- Ponder A., Najman, K.,Aninowski, M., Leszczyńska J., Głowacka, A., Bielarska, A.M., Lasinskas, M., Hallmann E. Polyphenols content, antioxidant properties and allergenic potency of organic and conventional blue honeysuckle berries, Molecules 2022, 27, 6083, 1-19.
Comment 7: “Page 2, Line number 72-77: This is okay. But the research gaps and novelty of the study is not properly established in the Introduction section. Author need to add one para reviewing the need of the study, and depicting the novelty of the study in the present day scenario as well as with its future approaches ”
Authors’ response: The Authors have emphatically demonstrated in lines 72-77 that there is absolutely no information in the literature about the comparison of organic and conventional edible flowers. If such information is missing, it means that the presented study is with a high novelty and fits perfectly into the trends of modern science and the demand for knowledge. Authors want to underline, that this is a preliminary experiment and will be continue in the future with a new aspects and scopes about quality of organic edible flowers.
Comment 8: “Results and Discussion: Page 4, Line number 115-116: This statement needs a perfect updated reference. ”
Authors’ response: According to Reviewer suggestion properly reference was add to manuscript text.
Comment 9: “ Page 4, Line number 117-118: There is no link between these two statements. Why?? Line number.”
Authors’ response: According to Reviewer suggestion, the incorrect sentence was corrected. It is now factually correct.
Comment 10: “ Page 4, Line number 128: Grammatical mistake. "It is confirm theory..." ”
Authors’ response: According to Reviewer suggestion incorrect sentence has been corrected in the text of the manuscript
Comment 11: “ Page 4, Line number 131: I want to see the results of reference number [63]. Spo that it will be easy for the readers to compare”
Authors’ response: According to Reviewer suggestion values of polyphenols concentration in organic and conventional pomegranate were added to manuscript text.
Comment 12: “Page 4, Line number 135: What are "Withe daisy flowers???" Is it white daisy flowers or what?? Need clarifications in this aspect. ”
Authors’ response: Authors want to apologise for mistake. Of course not “Withe daisy flowers” but White daisy flowers”. The mistake was corrected in manuscript text.
Comment 13: “Page 5, Line number 176-178: Where is the full stop "." for this statement?? ”
Authors’ response: Authors want to apologise. The long sentence has been divided into short and missing dots have been added.
Comment 14: “Page 5, Line number 178: "In our experiment".... You should write "In the present experiment".”
Authors’ response: According to this and previous Reviewer suggestions, in whole manuscript personation for Authors as: “our”, “I”, “we” were removed from manuscript text. Those words were change into more properly without personation.
Comment 15: “Page 6, Line number 213-214: What is conventional treatment?? ”
Authors’ response: Authors want to apologise for mistake. Of course not “conventional treatment??” only should be: “conventional pansy flowers”. This mistake was correct in manuscript text.
Comment 16: “Table: Table 2a: Are these words complete??? Something is missing, I think ”
Authors’ response: Table 2a heading was corrected.
Comment 17: “ Table 2a: Instead of writing "beta" author should use symbols. Follow the pattern throughout the whole manuscript.”
Authors’ response: According to Reviewer suggestion, the word “beta-“ has been replaced by the symbol β in the whole manuscript text.
Comment 18: “Page 10, Line number 353-355: It should not be like "Our results have been confirmed by others"... It should be your approach that you have confirmed and validated previously published results from this present investigation. Author needs to rewrite the statement with scientifically sound manner. ”
Authors’ response: Authors want to apologise for misunderstanding. According to Reviewer suggestion wrong grammatically sentence was corrected as well as substantively meaning is properly now.
Comment 19: “Figure 1: Author should compare this antioxidant activity with a standard antioxidative agent as well. ”
Authors’ response: Authors want to clarify that presented experiment is not methodological experiment, which purpose is to find a lot of methods analysis, solvents and validation of chemical methods. In our opinion used antioxidant activity measured by ABTS method was validate in the future and is no needed to use extra standard for showing antioxidant power. Of course Authors agree with Reviewer, that different antioxidants as anthocyanins and carotenoids have different antioxidative power. Additionally w manuscript text some information with properly references about different reactions of plant crops contained both carotenoids and anthocyanins and they reaction for ABTS were added.
Comment 20: “Materials and Methods: Page 12, Line number 409-410: Grammatical mistake.”
Authors’ response: According to the Reviewer suggestion, grammatical mistake was corrected in manuscript text now.
Comment 21: Page 12, Line number 411-412: By whom exactly the identification was carried out?? Need to mention the name with designation of the identifier.
Authors’ response: If the Authors understood the Reviewer's comment well, is it about identifying the flowers on the farm? Authors want to clarify. The farms’ owner did that in every farms identification of flowers species and types. In documents, the owners showed us certificate, guaranteed by seeds producers, what species and cultivar name is produce. It is according to organic and conventional plants production rules. In our opinion, it is not necessary to add this information to manuscript text.
Comment 22: “ Page 12, line number 412: What did the author meant by "extragently"??”
Authors’ response: Authors want to apologise. It should be not “extra gently” , but “a very delicate, to avoid flowers’ destruction”. This sentence was corrected in manuscript text now.
Comment 23: “ Page 12, Line number 423: What was the instrument used to store the samples at -80 Degree C??”
Authors’ response: Authors want to clarify: instrument used to store the samples at -80 oC was ultra-low freezer with temperature range keeper -1oC till -86oC, Serial number and name of producer were added into manuscript text.
Comment 24: “ Page 14, Line number 530: Will it really be "resolved"??”
Authors’ response: Authors want to apologise for mistake. Of course should be dissolved not resolved. Mistake was corrected into manuscript text.
Comment 25: “Page 14, Line number 547-548: Provide the formula in equation form. ”
Authors’ response: According to the Reviewer suggestion, equation for antioxidants activity calculation was added into manuscript text.
Comment 26: “ Conclusions: Page 15, Line number 563: Avoid the use of words such as our, we, I, you etc., throughout the manuscript.”
Authors’ response: This suggestion was corrected according to previous Reviewer remark.
Comment 27: “ Page 15, Line number 566-567: Statement is grammatically wrong.”
Authors’ response: According to Reviewer suggestion, wrong grammatically sentence was corrected in manuscript text now.
Comment 28: “ . Page 15, Line number 570-571: This cannot be recommended for every one right. Because toxicological site of consumption is still need to be evaluated.”
Authors’ response: Authors agree with Reviewer opinion. The last sentence of Conclusion section was corrected substantively.
Comment 29: “ References: All the references must be in accordance to the Journal format.”
Authors’ response: After carefully checking all references are in accordance with Plants journal format.
Comment 30: “ Additionally, the language of the manuscript must be revised with the help of a language expert. The revised version of the manuscript should must be devoid of any grammatical or typical errors.”
Authors’ response: Manuscript was revised by professional language expert. Below you can see language certificate.

Round 2
Reviewer 2 Report (Previous Reviewer 1)
Dear Authors
All the comments has been sussesfuly hence MS is in the acceptable form.
This manuscript is a resubmission of an earlier submission. The following is a list of the peer review reports and author responses from that submission.
Round 1
Reviewer 1 Report
I have reviewed this manuscript “plants-2224852-peer-review-v1”entitled, “Identification and quantification of bioactive compounds in organic and conventional edible pansy flowers (Viola × wittrockiana) and their biological activity”. The manuscript has tried to evaluate and show the differences in bioactive compound concentrations between different colour edible flowers of pansy. There are a few missing points and things that are needed to be corrected. The comments are given below for the author to make necessary changes and the manuscript can be accepted after major revision.
Reviewer’s comments:
1) In the keywords line no. 27, Write them in alphabetical orders.
2) In Introduction, you have given some notes on Chrysanthemum and peony which I don't think is that necessary. But there is no single description about the main plant i,e pansy.Write the morphological, distribution, uses etc. on Pansy flower.
3) In table 2, it should be flower colours.
4) In material and methods, line no 301 and 302, it should be grinded.
5) In Picture 1, changes ‘flowes’ to flowers, ‘violer’ to violet.
6) In line no 338, it should be written as F.W.
7) In conclusion, write the conclusion more longer and follow a uniform format and size with the rest of the manuscript.
8) In references, arrange them in alphabetical orders.

Reviewer 2 Report
Thank you for allowing me to review this interesting article which aims to study the differences in the composition of bioactive elements in organic and conventional pansy flowers according to their color. This fascinating and well-written article meets IERPH standards for publication in the journal. However, several considerations should be taken into account by authors before publication:
1. Lines 37, 59, and 68: Pro-health is ambiguous. I propose changing it to another term, such as "beneficial effects on human health" or similar. Otherwise, I would ask the authors to define what they mean by pro-health.
2. The materials and methods section should precede the results and discussion section. It would make it easier to read, according to a more logical order.
3. In the "flower origins" section, what conditions were the flowers transported? If fresh samples were stored until analysis, what were the conditions under which this storage occurred?
4. Line 391: please check the typo.
5. Supplementary file: I do not know if it is a problem with my computer, but the supplementary material file cannot be displayed correctly. I would be grateful if this could be checked.
Reviewer 3 Report
Dear Author,
Firstly, Identification and quantification, as mentioned in the manuscript title but really, I didn’t understand how you perform the identification and the quantification of the compounds, it need more explanation in result section. Also, as mentioned in material and methods section: Identification and quantification of flavonoids, phenolic acids, carotenoids, and polyphenols were performed according to standards and spectra: it’s not clear it needs more explanation. I expected to see HPLC chromatograms for samples as well as for standard molecules?
Additionally, you mentioned in the title : biological activity, but in the manuscript there is only antioxidant activity?
As mentioned in abstract section, all experiments were done using one technic the HPLC-DAD method, I think it’s not enough. Also for the supplementary data its not clear it need more explanation (spectra of what?)
In introduction section you give details on biological activities of flowers, but no experiment was done to show these benefic pro-healthy properties
Result and discussion section:
Table 1 and 2: I got confused, for the production system (organic or conventional) it’s not clear, which variety (flowers color) you used for organic and which one you used for conventional?
I strongly suggest to add explanations about Identification and quantification of compounds (added HPLC chromatograms for samples as well for standard molecules,….. )
I suggest conducting some biological activities to demonstrate the presence of potent active compounds in the extracts and show the difference between the used production systems.